# A Bayesian framework for the detection of diffusive heterogeneity

**Julie A. Cass**[1]*, **C. David Williams**[1], **Julie Theriot**[1,2]

**1** Allen Institute for Cell Science, Seattle, WA, United States of America, **2** Department of Biology, University of Washington, Seattle, WA, United States of America

* juliec@alleninstitute.org

## Abstract

Cells are crowded and spatially heterogeneous, complicating the transport of organelles, proteins and other substrates. One aspect of this complex physical environment, the mobility of passively transported substrates, can be quantitatively characterized by the diffusion coefficient: a descriptor of how rapidly substrates will diffuse in the cell, dependent on their size and effective local viscosity. The spatial dependence of diffusivity is challenging to quantitatively characterize, because temporally and spatially finite observations offer limited information about a spatially varying stochastic process. We present a Bayesian framework that estimates diffusion coefficients from single particle trajectories, and predicts our ability to distinguish differences in diffusion coefficient estimates, conditional on how much they differ and the amount of data collected. This framework is packaged into a public software repository, including a tutorial Jupyter notebook demonstrating implementation of our method for diffusivity estimation, analysis of sources of uncertainty estimation, and visualization of all results. This estimation and uncertainty analysis allows our framework to be used as a guide in experimental design of diffusivity assays.

**Data Availability Statement:** All data and software are publicly available on the GitHub repository https://github.com/AllenCellModeling/diffusive_distinguishability (DOI: 10.5281/zenodo.2662552).

**Funding:** This work has been completed and funded by the Allen Institute for Cell Science.

## Introduction

Diffusion is essential for the intra-cellular transport of many organelles, proteins and substrates. In the crowded and heterogeneous physical environment of the cell, diffusivity is a local, spatially dependent characteristic of the space, dependent on factors such as the size of the particle, and the local viscosity and spatial crowding. These spatial heterogeneities must be addressed when using diffusion coefficients as readouts of intra-cellular transport and the physical environment. This intra-cellular diffusion coefficient is often experimentally estimated through two approaches: single particle tracking (SPT) [1–3] and fluorescence correlation spectroscopy (FCS) [4].

In single particle tracking experiments, a live cell is imaged in successive frames, and individual punctate objects are tracked to construct a trajectory of time-dependent positions (Fig 1). One of the most common approaches to extracting diffusion coefficient estimates from SPT is to use mean-squared displacement (MSD). The MSD generically follows the following

**Competing interests:** The authors have declared that no competing interests exist.

relationship:

$$MSD(\tau) = \langle(\Delta x(\tau))^2\rangle = 2dD\tau^{\alpha},$$

(1)

where $\Delta x$ is the step size between frames taken at a time lag of $\tau$, in $d$ spatial dimensions, and $D$ is the diffusion coefficient. The parameter setting the MSD scaling with time, $\alpha$, is determined by the diffusive model. Any temporal scaling with $\alpha \neq 1$ is called anomalous diffusion, with super- and sub-diffusion models having $\alpha > 1$ and $\alpha < 1$, respectively. Intracellular diffusion has most often been characterized to be sub-diffusive, likely as a result of crowding [3].

For objects undergoing homogeneous isotropic diffusion, the MSD of puncta is a linear function of lag time ($\alpha = 1$), with the slope being proportional to the apparent diffusion coefficient: The averaging in this calculation can be taken on a single or multiple trajectory basis (i.e. mean of each displacement over time-step $\tau$ in a single trajectory or over many trajectories). If MSD analysis is completed on a per-trajectory basis, this technique allows for spatial resolution of diffusivity variation; however it relies on the fitting of the $MSD(\tau)$ slope. This analysis can be misleading, as it includes no information about the uncertainty in this estimation beyond calculation of the error on the mean. As a result, when multiple single-trajectory MSD's are plotted together on a log-log plot, it can be easy to interpret non-overlapping $MSD(\tau)$ line as portraying distinct diffusivities, when they could just be representing uncertainty-driven variations around a single shared value.

In FCS, a laser illuminates a region of a sample containing fluorescently tagged particles [5]. The characteristic time a fluorescent particle spends in the illuminated region ("dwell time") can be calculated from the intensity auto-correlation function. Together with the length scale of the illuminated region, dwell time gives an estimate of the diffusion coefficient in this region. The calculation of the diffusion coefficient from these properties is dependent on the chosen diffusion model; this method is flexible to anomalous diffusion models and captures small-scale local diffusivities. However, only one local measurement can be made from each illuminated region, making the assessment of many local regions experimentally intensive.

Like FCS, SPT can be used to probe local diffusivities and is robust to anomalous diffusion models [6]. But in contrast, rather than providing one diffusivity measurement per illuminated

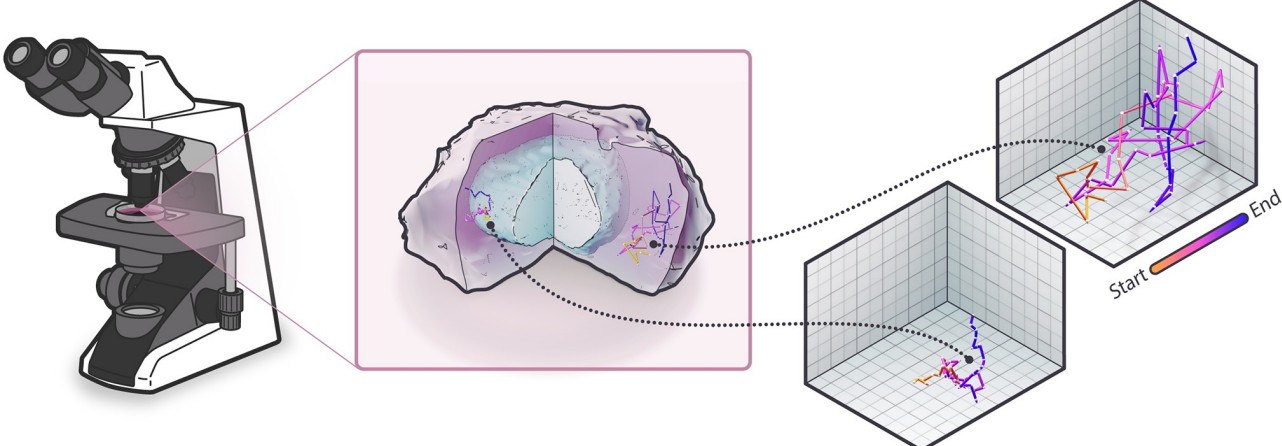

**Fig 1. Single particle tracking.** In SPT, a live cell is imaged over a series of time points. Individual punctate objects are localized at each time-step, and these positions are traced from frame to frame to produce individual time-lapse trajectories.

region, SPT allows for as many individual local diffusivity estimates to be simultaneously made as there are fluorescent particles in the field of view. Dependent on particle density, this advantage allows for the efficient use of spatially dependent diffusivity assays. While SPT offers many advantages, it relies on finite observations of a stochastic assay, limiting our diffusivity estimation accuracy.

While powerful analyses from SPT have indicated the complexity of transport in live cells, the spatial variation of the diffusion coefficient remains poorly characterized. This can be attributed to challenges in disentangling effects of biological heterogeneity and limited sampling of a stochastic process [7, 8]. To address these challenges, we developed a Bayesian framework to estimate a posterior distribution of the possible diffusion coefficients underlying single-trajectory dynamics. This framework generates look-up tables predicting the detectability of differences in diffusion coefficients, conditional on the ratio of their values and amount of trajectory data collected.

Other packages with information theoretic frameworks for trajectory analysis have been released; for example, the Single-Molecule Analysis by Unsupervised Gibbs sampling ("SMAUG") software package [9] also uses Bayesian estimation to characterize diffusive environments. However, our package is unique because it is intended specifically to provide lightweight trajectory analysis and prediction that can be used by those with a biological background to inform microscopy experiment design, without requiring deep statistical or computational knowledge.

## Materials and methods

### Trajectory simulation and localization error

We generated sample trajectories with known diffusion coefficients by simulating Brownian motion of particles in a d-dimensional space. At each time-point and along each spatial dimension, a step size was drawn from a zero-mean Gaussian $\mathcal{N}(\mu = 0, \sigma^2)$ with variance $\sigma^2$ defined by the diffusion coefficient: $\sigma^2 = \langle |\Delta x|^2 \rangle = 2dD\Delta t$, where $d$ is the number of spatial dimensions, $D$ is the homogeneous isotropic diffusion coefficient, and $\Delta t$ is the time-step. At each time point, a new step size in each dimension was drawn from the normal distribution, to generate the displacement vector $\vec{\Delta x}$. This displacement vector was added to the position $\vec{x}(t)$ to generate the next position $\vec{x}(t + \Delta t)$. We recorded the position of the particle at each frame in a time-series, constructing a trajectory mimicking the data one would get from tracking an object from time-series images (Fig 2).

To mimic the static localization error inherent in microscopy-generated trajectories in our simulated trajectories, we added Gaussian error to the locations of simulated particles at each time point [10]. After each successive location was stochastically chosen based on a model of Brownian motion, an additional draw from another normal distribution was made to select a shift in position in each spatial dimension. The variance of this Gaussian localization error can be tuned to the user's own specific microscope configuration.

The locations of the simulated particle at each time-point (with and without error included) are stored in a DataFrame, and these trajectories are digested into frame-to-frame displacements; realistically these step sizes were *used* to generate the trajectories, making back-calculating them seem tedious. However, the remainder of our toolkit is designed for analysis of any trajectory—simulated or tracked from images. Therefore a user can choose to either input their own image-derived trajectories or use a simulated trajectory to perform estimation of the unknown diffusivity.

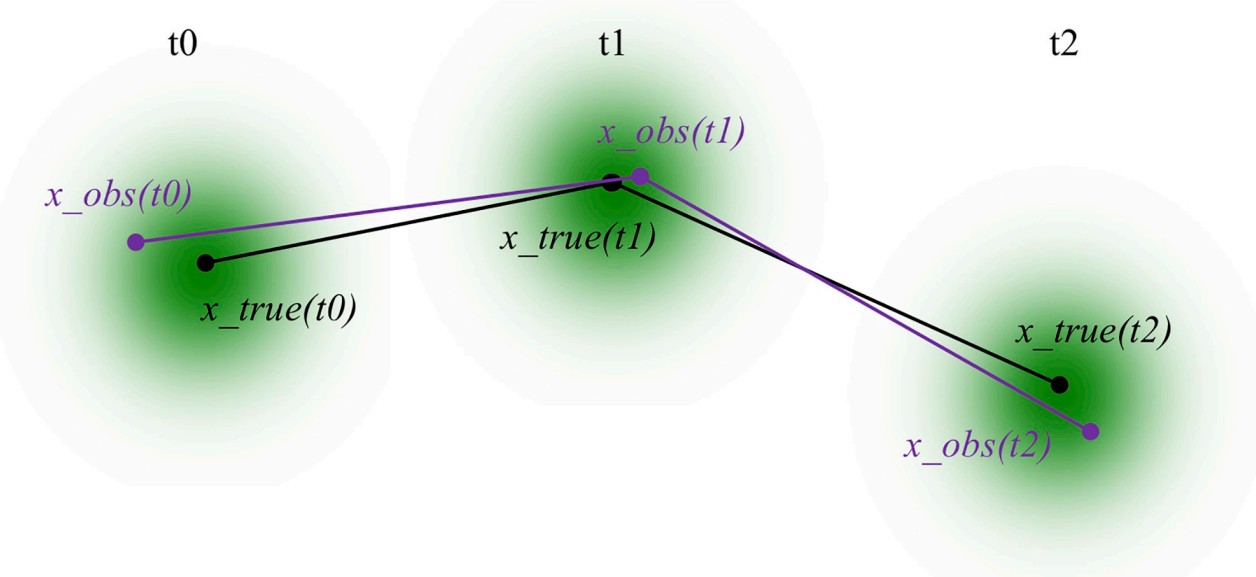

**Fig 2. Sample trajectory with and without localization error.** A 2D diffusive trajectory with no localization error is drawn for T time-steps. At each time-step, a cloud of Gaussian uncertainty is drawn; the shape and shading of this cloud demonstrate how likely it is for the position of be measured at any of the surrounding points rather than in the true position. A sample alternative trajectory is drawn (purple) showing the path we might observe the particle to take, due to the localization error in measuring the true position as a function of time.

## Bayesian inference of diffusivity

To estimate the diffusivity underlying a single trajectory (and our uncertainty in this estimation), we employ Bayesian inference [11]. This method is focused on generating a "posterior probability distribution": the probability that a random variable takes on any of a set of values, based on provided evidence and a prior distribution. In our case, the random variable is the diffusivity, and the evidence is the set of step sizes from a single trajectory. The prior distribution for the variance of a normal distribution with known mean is an inverse-gamma distribution. This acts as a conjugate prior; that is, a class of distributions for which the prior and posterior distributions take on the same mathematical form; therefor our posterior will also be an inverse-gamma function. The inverse-gamma distribution's probability density function over diffusion coefficients $D > 0$ is parameterized by the scale ($a$) and shape ($b$):

$$IG(D; a, b) = \frac{b^a}{\Gamma(a)} (1/D)^{(a+1)} e^{-b/D}. \tag{2}$$

The parameters $a$ and $b$ have been used in place of the typical use of $\alpha$ and $\beta$ respectively, to disambiguate from the MSD time-scaling parameter $\alpha$ in Eq 1.

The posterior distribution peaks near the true diffusion coefficient and has a width corresponding to the confidence interval of our estimate, which is largely determined by the trajectory length and magnitude of localization error.

## Characterizing the distinguishability of diffusivity posteriors

To characterize our uncertainty on whether trajectories come from regions with different diffusivities, we require a way to quantitatively discriminate between pairs of posterior distributions. To achieve this, we use the Kullback-Leibler (KL) divergence. The KL divergence acts as a single-value estimation of how well we can analytically distinguish whether the step sizes

from a trajectory came from the diffusivity predicted by one posterior or the other. The KL divergence of two inverse-gamma distributions $p(a, b)$ and $q(\hat{a}, \hat{b})$ is calculated as follows [12]:

$$KL(a, b, \hat{a}, \hat{b}) = (a - \hat{a})\Psi(a) + \hat{b}\left(\frac{a}{b}\right) - a + log\frac{b^{\hat{a}+1}\Gamma(\hat{a})}{b\hat{b}^{\hat{a}}\Gamma(a)} \tag{3}$$

where $\Psi(a)$ is the digamma function, defined as the logarithmic derivative of the gamma function ($\Gamma(a)$). Since this metric is not symmetric and we have no preference between distributions $p$ and $q$, we use a symmetrized version of the KL divergence

$KL = \frac{1}{2}\left(KL(a, b, \hat{a}, \hat{b}) + KL(\hat{a}, \hat{b}, a, b)\right)$.

## Code availability

A repository for our source code is publicly available at the Allen Cell Modeling GitHub page https://github.com/AllenCellModeling/diffusive_distinguishability, conveniently packaged with ReadTheDocs documentation and a tutorial Jupyter notebook demonstrating usage and reproducible figure production. This package is registered under DOI 10.5281/zenodo.2662552.

## Results and discussion

### Bayesian inference of diffusivity

When the position of a diffusing object is recorded as a trajectory of discrete steps in time, the sizes of those steps can be mathematically represented as stochastic draws from a distribution characterized by the diffusion coefficient. Our method for estimating the diffusion coefficient relies on breaking individual trajectories into frame-to-frame steps, and applying a Bayesian statistical framework to predict the diffusivity underlying each set of stochastically derived step sizes. From a single trajectory, this framework provides not only an estimation of the diffusivity, but also a representation of our uncertainty. While our framework could be adapted to analyze more complex dynamic models, our current implementation introduces a workflow for analyzing isotropic homogeneous diffusion; therefore, trajectories with unknown diffusivity will result in a step-size distribution which is normally distributed, with zero mean and unknown variance $\mathcal{N}(\mu = 0, \ \sigma^2)$.

Bayesian inference is built on the use prior and posterior distributions [11]. Our "prior" distribution is an initial guess at the solution to a problem before using our observations or data to inform our expectations (i.e. *a priori*); for instance, if I have no intuition for the solution to my estimation problem, I would use a flat prior telling my model that I think any solution is equally likely. We then use our data to narrow down our solution estimation (i.e. *a posteriori*), resulting in a "posterior" distribution. In our case, the step size distribution from a single trajectory would be the observations, and the posterior might look like a distribution of diffusivity values, peaked around some value indicating a likely estimate of the underlying diffusion coefficient. The longer the trajectory is, the more information we can use to narrow down our answer, leading to a more tightly peaked posterior (discussed in greater detail in the *Sources of posterior estimate error*).

### Inverse-gamma distribution as diffusivity conjugate prior

In this section, we will step through the process of applying Bayesian analysis to our particular case. First, we will get introduced to the governing principle of this approach, called Bayes'

theorem [11], then we will carefully digest this principle into pieces and see how it applies to our own application.

Bayes' theorem tells us that the posterior distribution for an unknown variable $\theta$ is proportional to the product of the prior distribution p($\theta$) and the "likelihood function", or the function giving the probability of making observation $x$ given the unknown variable $p(x|\theta)$. Mathematically, this is often represented:

$$p(\theta|x) \propto p(\theta)p(x|\theta). \tag{4}$$

How does this apply to the diffusion process we have been exploring? In our problem, we have taken single particle trajectories and split them into frame-to-frame step sizes. We can say, then, that our Bayesian "observed variable $x$" is the step size $\Delta x$. We've discussed previously that we expect the step sizes for diffusive trajectories to be normally distributed, with a mean of zero and an unknown variance. Translating again to the Bayesian framework, we can say that our unknown variable $\theta$ is the variance $\sigma^2$, and our likelihood function is the normal distribution of step sizes, i.e. $p(x|\theta) = p(\Delta x|\sigma^2) = \mathcal{N}(0, \sigma^2)$.

The prior is our initial guess of the probability distribution of values for our unknown variable, $\sigma^2$. To determine the prior distribution for our cases, $p(\theta) = p(\sigma^2)$, we consider the mathematical dependence of the normally distributed step sizes on the variance $\sigma^2$:

$$p(\Delta x|\sigma^2) \propto (1/\sigma^2)^{\beta} e^{-\gamma/\sigma^2} \tag{5}$$

We see that this dependence looks a bit like a gamma distribution, except that our variable of interest is found in the denominator. This class of function is intuitively called an inverse-gamma function (*IG*, Eq 2). We can now say *a priori* that we expect our estimated $\sigma^2$ values to follow an inverse-gamma distribution, and therefore this is the form of our prior: $p(\theta) = p(\sigma^2) = IG(\sigma^2)$.

We have now seen how to place the observed and unknown Bayesian variables in the context of our problem, and explored the Normal and inverse gamma distributions which can be used as our likelihood and prior distributions, respectively. With these pieces in hand, we can now find the class of function for our posterior distribution, as the product of our prior and likelihood distributions (Eq 4). In our case, we find that the product of $p(\sigma^2)$ and $p(\Delta x|\sigma^2)$ also has an inverse gamma dependence on $\sigma^2$. We note that our posterior distribution is a function of the same class as the posterior—we will come back to this after a brief note.

In this section we have built up a framework for performing Bayesian analysis to estimate a distribution of variances, but we promised an estimation of the diffusion coefficient. Now let us recall that the variance of the diffusive step size distribution is directly proportional t the diffusion coefficient ($\sigma^2 = 2dD\Delta t$), and therefore, with the inclusion of a multiplicative constant, this analysis is easily transferred into a Bayesian estimation of diffusivity $D$, with inverse gamma prior and posterior distributions $IG(D)$.

In general, when the prior and posterior for Bayesian analysis take the same mathematical form, the prior is referred to as a "conjugate prior." The matching of the conjugate prior and posterior function types dramatically simplifies the statistical method, presenting one advantage of this prior. A second advantage of our prior is that the inverse-gamma distribution acts a conservative initial "guess," with any order of magnitude diffusivity is equally likely, before the introduction of any data. In the Bayesian method of statistical inference, the choice of prior can bias our results; for instance, if we expect the diffusivity to be around 1 $\mu m^2/s$, we might select a prior distribution that is sharply peaked around this value. If the diffusivity is, in fact, close to this value, that choice of prior would help guide our posterior towards the correct result. However, if that intuition is incorrect, and the true value lies in the tails outside our

peaked prior, we will have biased out Bayesian estimator away from the true value, skewing our results ti be incorrect. As a result, use of an "uninformative" prior such as the inverse-gamma distribution with scale and shape parameter $a$, $b \rightarrow 0$, treats all posterior results as being equally likely and helps us to remove our *a priori* bias from our diffusivity inference. The distribution and quantity of values in our set of step sizes will then determine the scale ($a$) and shape ($b$) parameters for our posterior inverse-gamma distribution $IG(D;a, b)$.

## Sources of posterior estimate error

The estimation of diffusivity from a single trajectory is limited by the finite trajectory length and accuracy in localizing the object at each time point. As a result, careful consideration of how each of these factors will impact the estimation uncertainty is necessary when constructing an experimental design. To address this, we have constructed a framework for generating look-up tables predicting the percent error posterior diffusivity estimation conditional on a set of trajectory lengths and localization errors.

Many methods for estimating diffusivity from a single trajectory rely on the analysis of the frame-to-frame step-size distribution extracted from that trajectory. However, during a microscopy experiment, there will always be an inherent limitation to the degree of accuracy that an object can be localized in each frame. This arises from both static and dynamic sources of localization error; static localization error occurs due to the inherent limit to spatial resolution of imaging experiments, while dynamic localization error comes from the non-instantaneous nature of capturing an image resulting in object movement during image acquisition [13]. Since dynamic localization error is most relevant for quicky moving objects, such as small substrates, we have chosen to simulate and provide example analysis of the effects of static localization error.

As a result of limitations in spatial resolution, when the object is tracked and trajectories generated, an inherent limitation in localization accuracy is encoded in the trajectory, and therefore skews the step-size values being used to infer the diffusion coefficient. To demonstrate the impact of localization error on SPT, we provide an example simulated trajectory with varying amounts of localization error applied (Fig 3).

Fig 4 demonstrates the impact of underlying diffusion coefficients and localization errors on posterior estimates. We provide examples of trajectories in two regions with differing diffusion coefficients, each with and without localization error included in the trajectory simulation. We then plot the posteriors for all four of these trajectories on one set of axes. Our tool aims to quantify the effects of this localization error on the estimation of diffusivity by generating trajectories with varying known degrees of localization error and reporting their impact on the error of the posterior estimation of the known underlying diffusivity.

Diffusive trajectories are composed of successive steps, whose sizes are stochastic draws from a distribution set by the diffusivity. When only short trajectories are available, we have only a limited set of draws from this distribution—as a result, the variance of this distribution is difficult to accurately predict, and the posterior distribution of diffusivity probabilities will be less accurate and precise. While it would be ideal to simply collect longer trajectories, this is often experimentally impossible; therefore, we aim to give experimentalists an analysis framework to estimate how accurately they can predict diffusivity given their own limitations in tracking.

Because our trajectories are simulated, we benefit from the knowledge of the true diffusivity and degree of localization error, and can therefore precisely quantify the relation between the error in our Bayesian estimation of diffusivity and the level of localization error. This provides

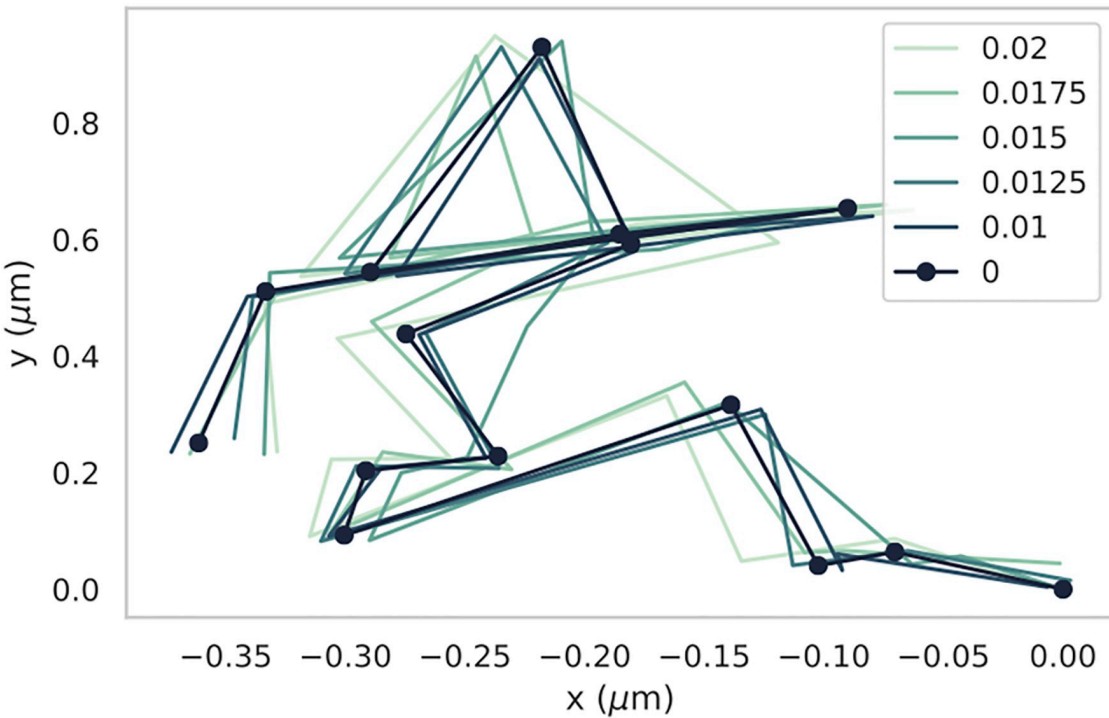

**Fig 3. Sample trajectory with and without localization error.** A 2D diffusive trajectory with no localization error is drawn for T time-steps. That same trajectory is then redrawn in increasingly light colors, for increasing levels of localization error. This error is parameterized in the form of the standard deviation of a Gaussian blur, in microns. This example allows us to visualize the impact that a range of localization errors would have on the same trajectory.

a look-up table for experimentalists to predict the accuracy in diffusivity estimation that can be achieved with their own particular microscopy experiment, shown in Fig 4. We quantify the error in our estimates as the magnitude of the percent error between the true diffusivity and the mode of the posterior probability distribution as calculated by the posterior's scale and

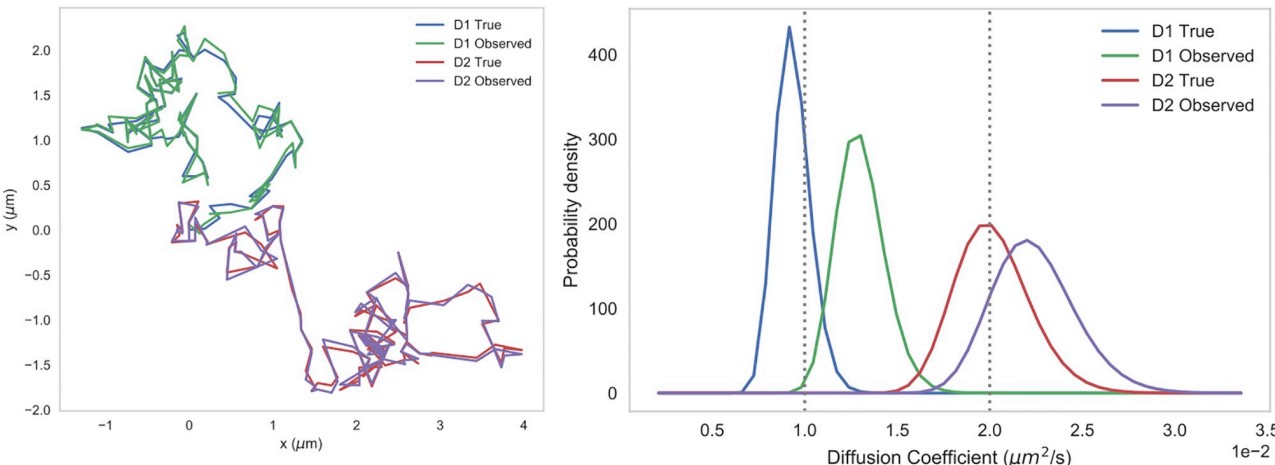

**Fig 4. Sample trajectories and diffusivity posteriors, with and without localization error.** Left: Sample simulated 2D trajectories composed of 100 steps with diffusion coefficient $D_1 = 0.01 \ \mu m^2/s$ and $D_2 = 0.02 \ \mu m^2/s$. The "Observed" trajectories are generated with localization error 0.05 $\mu m$, while the "True" trajectories have no localization error. Right: Posterior distributions for all trajectories. These posteriors are all inverse-gamma distributions generated using our Bayesian inference framework.

shape parameters:

$$\%_{error} = |100\left(\frac{b+1}{a} - D_{true}\right)/D_{true}|. \tag{6}$$

Of course, due to the stochastic nature of diffusive properties, even with all the same simulation parameters, the posterior error will vary from one simulation to the next. In order to capture the mean effect of each parameter on posterior error, the results in Fig 5 represent the average percent error for $N = 10^4$ replicates of the same simulation parameterization.

For example, in a study of the *Bacilus subtilis* SMC complex [3], with diffusion coefficient on the order of 0.1 $\mu m^2/s$, localization error of 0.1 $\mu m$ and trajectory lengths of approximately 50 frames, this table tells us to expect a diffusivity estimation error of $\approx 15\%$. A study of MRNP diffusion in the nucleus [14] with diffusion coefficient also on the order of 0.1 $\mu m^2/s$, but with localization errors ranging from 0.01-0.1 $\mu m$ and trajectory lengths greater than 1000 frames, we can expect a diffusivity error ranging between 5% and 10%, depending on the experiment's localization error. For a more specific deep dive into the estimation error expected for a specific diffusivity, localization error and trajectory length, users can simulate these results using the "get_dim_error" function of our tool, demonstrated in the Jupyter notebook tutorial.

In addition, it should be noted that the number of spatial dimensions of the assay (i.e. whether trajectories are measured in two or three spatial dimensions) as well as the mean-squared displacement (related to the diffusion coefficient) can impact the relationship between localization error and Bayesian estimation error. For a more in-depth discussion and simulation of this, please see the tutorial Jupyter notebook in our project GitHub repository.

## Distinguishability of trajectory diffusivities

With the above percent error analysis derived for simulated trajectories with known diffusivities, a picture arises of how our estimates of the diffusivity differ from the true values. As a result, when this technique is applied to experimentally-derived trajectories whose underlying diffusivities are unknown, we may want to ask 'how likely is it that two trajectories resulting in different diffusivity estimates were actually derived from regions with the same diffusivity?' The biological motivation and analog for this technical question is 'how heterogeneous is the physical cellular environment?'

This will depend on the amount of overlap between the two diffusivity posterior distributions, which is determined by: (1) how different the underlying diffusion coefficients are (how far apart the theoretical maxima of posteriors are) and (2) how uncertain we are in our estimations (how wide the posterior distributions are). One way to measure the difference between two distributions is to use the Kullback-Leibler divergence (KL divergence). A KL divergence of zero indicates that two distributions are identical; one interpretation of this metric is that its inverse tells you the number of times you can draw samples from one distribution in place of the other before there is significant information loss.

In order to communicate the distinguishability of pairs of posteriors conditional on their trajectory parameters, we have created a heatmap look-up table of the KL divergence of posterior pairs, dependent upon the ratio of their underlying diffusion coefficients (i.e. $D_2/D_1$), and the trajectory length. An example of this look-up table heatmap is provided in Fig 6. The complete code used to generate this map is provided in the tutorial Jupyter notebook found in the GitHub repository for this project. By cloning the repository, users can directly edit this example code to recreate this map with a different localization error or different distribution of

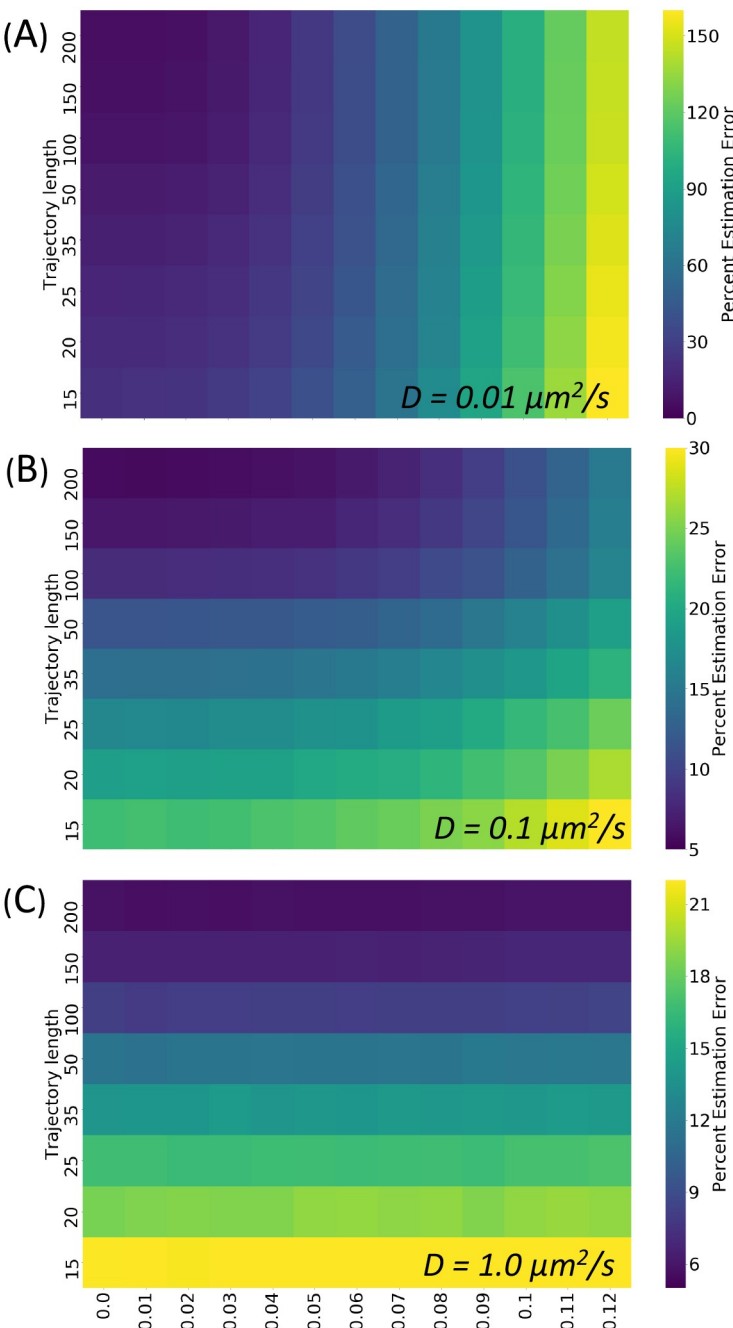

**Fig 5. Percent posterior estimation error conditional on static localization accuracy and trajectory lengths.** The percent error for a given posterior is measured as the percent error between the true diffusion coefficient used to generate the trajectory, and the mode of the posterior distribution (or the diffusion coefficient which gives the maximum value of the probability density function). This heatmap reports the mean percent error magnitude for $10^4$ posteriors generated under each set of trajectory length and localization error conditions, with diffusion coefficients of (A) 0.01 $\mu m^2/s$ (B) 0.1 $\mu m^2/s$ and (C) 1.0 $\mu m^2/s$. Please note the difference in heatmap scale bars.

trajectory lengths and diffusion coefficient values. An experimentalist may generate their own heatmap for trajectories with their specified degree of localization error, and get a table to tell them how distinguishable differences in diffusion coefficients will be for different lengths of trajectories that they can collect. This framework could therefore play a valuable role in

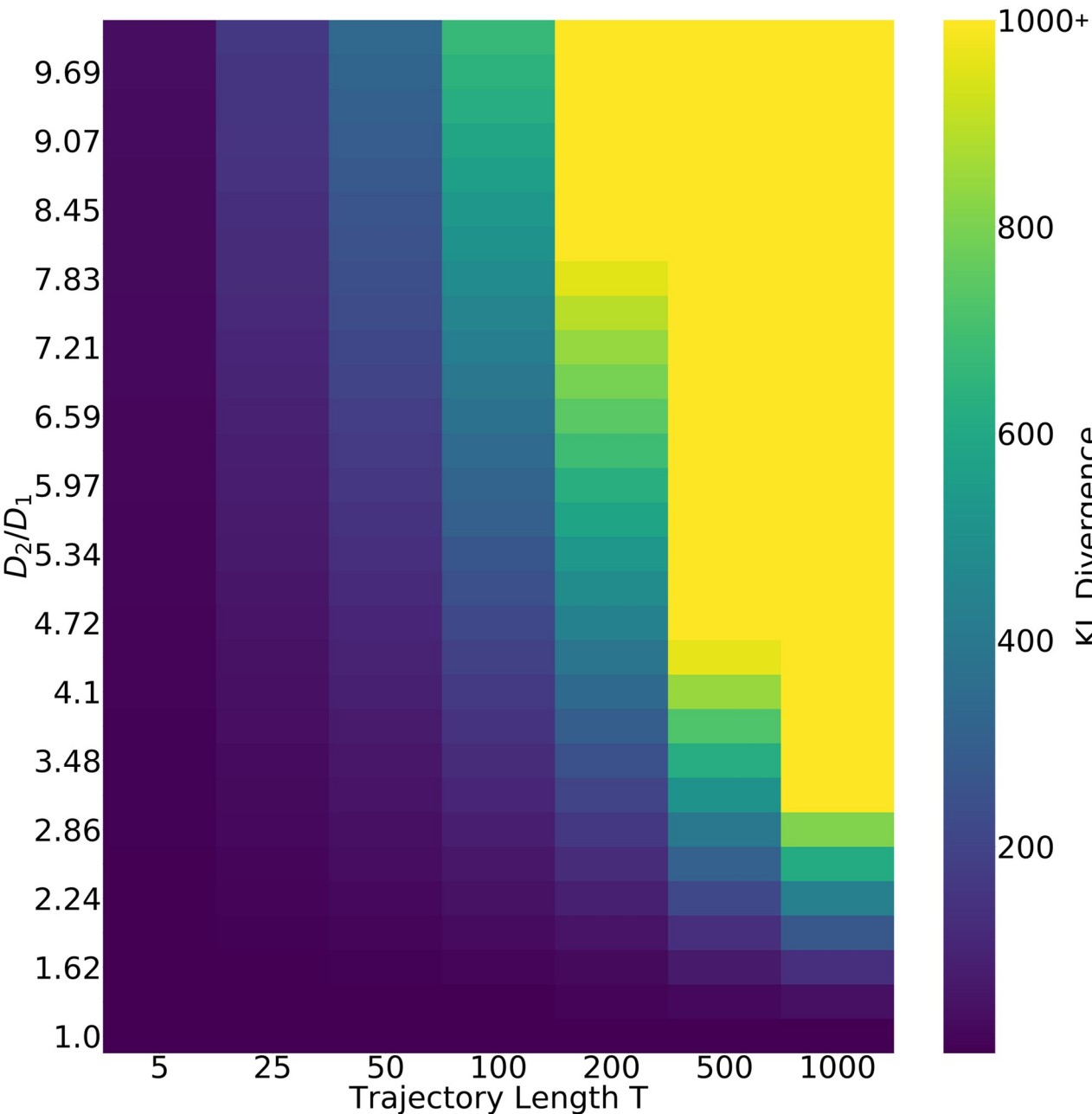

**Fig 6. Look-up table for posterior KL divergence, conditional on diffusivities and trajectory lengths.** Heatmap displaying the average KL divergence of diffusivity posteriors. For each entry in the heatmap, two trajectories of the same length (x-axis) are produced, with differing underlying diffusivities with the ratio $D_2/D_1$ (y-axis). A posterior is estimated for each, and their KL divergence is calculated as a measure of the distinguishability of the underlying diffusivities. As this process is stochastic, this is repeated $10^4$, with the average being the value reported in the heatmap.

describing the feasibility of and requirements for experiments addressing the spatial heterogeneity of the intra-cellular diffusive environment.

## Comparison with MSD analysis

Given a single trajectory, let us compare what we could learn of the underlying diffusivity through MSD analysis and our Bayesian framework. In MSD analysis, the trajectory would be

split into step sizes associated with every possible lag time (that is, the mean of the squared displacement for all step sizes between frames $\tau$ = 1, 2, 3. . . frames apart. The diffusivity can be calculated by fitting the MSD using Eq 1, often using a loglog plot. This provides a single prediction of the average diffusivity over the course of the trajectory. In contrast, our Bayesian framework outputs a probability distribution of diffusivity values; the diffusivity giving the highest probability can be extracted to give a single-values diffusivity estimation, but the distribution as a whole offers the appealing advantage of giving a quantitative measure of our confidence in this estimate.

This confidence interval offers an added benefits over MSD analysis. Through posterior visualization and the KL divergence analysis described in the previous section, this Bayesian estimation framework provides us with a straightforward visual and quantitative way to diagnose how likely it is that diffusivity estimates from two trajectories are actually describing regions with different physical properties. In the case of MSD, comparison of single-trajectory diffusivity estimates is done by plotting $MSD(\tau)$ for each trajectory on the same log-log plot and comparing their intercepts. This methodology fails to capture information about uncertainty, and may lead to the false conclusion that each trajectory is taken from a region with a unique diffusivity. In many cases Bayesian posterior analysis will reveal significant overlap between these trajectories' posteriors, indicating the analyzed trajectories do not mark the region as having heterogeneous diffusivity. One interpretation of the KL divergence is that its inverse tells you the number of observations you can make using one distribution in place of the other, before the information loss becomes significant. For instance, if posteriors from trajectories A and B have a KL divergence of 0.01, I could use 100 measurements from posterior A to describe posterior B before I start to significantly misinterpret posterior B; this means that these distributions are extremely similar and their diffusivities might be considered to be the same. If posteriors A and B have a KL divergence greater than one, the numbed of observations before significant information loss would be less than or equal to one, telling me that using even a single measurement from one distribution in place of the other will cause a mischaracterization; the trajectories used to generate these distributions have distinct diffusivities.

## Application to spatially dependent diffusivity characterization

In the introduction of this paper, we discussed the importance of analysis techniques that acknowledge the heterogeneity of cellular environments. The single-trajectory dependence of this tool offers a framework to build on for characterizing variations in the diffusivities felt by trajectories recorded in different cellular regions. By mapping the diffusivity estimates from each trajectory (value most probable from posterior distribution) to the spatial region where the tracked substrate was localized, the user can build up a spatial mapping of the diffusivity. While frameworks exist for spatial mapping of the physical properties of cells, such as nanorheology of injected particles [15] and SMAUG [9], these techniques respectively require an extensive and invasive experimental design or in-depth knowledge of computational Bayesian inference. Our tool offers an approachable framework for experimental design of studies to probe the spatial variation of physical properties of the cell.

## Framework limitations

As we have discussed, the presence of localization error and the finite nature of trajectories will contribute to the uncertainty in any analysis of single particle trajectories. Here, we discuss several other important limitations to be considered when using this software package.

This framework is currently only implemented for the analysis of pure diffusion, however anomalous diffusion (particularly sub-diffusion) is commonly reported in the analysis of

biological trajectories. Users could adapt the package to analyze trajectories undergoing anomalous diffusion by editing our Bayesian estimation code. We have described how our conjugate prior and posterior model have been selected specifically to analyze a normal distribution of step sizes with zero mean; because the step size distribution is dependent upon the diffusion model, the class of function used for the prior and posterior will also be dependent upon the diffusion model. To modify this framework for other diffusion models, users would therefore select new prior and posterior distributions, and require a new equation for calculating the KL divergence for a pair of distributions belonging to this mathematical function class (i.e. a replacement for Eq 3). However, it is important to note that as the diffusion model becomes more complex, selection of a prior and posterior can become very challenging, limiting the scope of the framework.

Realistic intra-cellular transport is additionally complicated by the presence of active transport and flow. Furthermore, the affects of confinement and characterization of the physical properties of the cytoplasm (i.e. elasticity) can further complicate intra-cellular dynamics. As these factors are not considered in the current implementation of our framework, they will contribute to the error in the analysis of experimentally derived trajectories.

### Application to fractional Brownian motion trajectories

Many research studies have demonstrated intracellular transport to be sub-diffusive (i.e. $\alpha < 1$ in Eq 1), with $\alpha = 0.75$ in crowded cellular environments such as an actin lattice or the cytoplasm [3]. In particular, these trajectories have ergodic MSD's and velocity autocorrelations which are anti-correlated at short timescales; this behavior is characterized by the sub-diffusive model of fractional Brownian motion (FBM). FBM trajectories are parameterized by the Hurst coefficient H, defined as $H = \alpha/2$; thus, FBM trajectories with $H = 0.375$ provide a more complicated, but more realistic representation of intracellular transport than the simpler model of pure diffusion used in the results presented so far. However, its application in this Bayesian estimation tool would require the use of a much more complicated prior and posterior; while our tool is built to be robust to varying priors and posteriors, we understand that defining these distributions for more complex models of motion can be challenging. To test how accurately this simpler diffusion model can be used to predict the diffusivity of more realistic FBM trajectories, we applied the existing, pure-diffusion based Bayesian analysis to FBM-generated trajectories with $H = 0.375$ and calculate the error in the estimated diffusivity, presenting the resulting posterior error heat-maps as in Fig 7. These FBM trajectories are produced using publicly available simulator written by Christopher Flynn (https://pypi.org/project/fbm/). It is important to note that for this sub-diffusive behavior, there is no longer a single diffusion coefficient defined for the trajectory; instead, the diffusion coefficient must now be defined for a given time lag ($\tau$ in Eq 1). For the results presented here, we analyze the error in the effective diffusion coefficient defined for $\alpha = 1$ second. We find that the error in the estimated diffusivity for these more biologically relevant trajectories are nearly identical to those reported for the purely diffusive trajectories; we therefore believe that despite that complexity of experimentally derived intracellular trajectories, this analysis tool remains a suitable for experimental diffusivity estimation.

### Conclusion

Heterogeneity of diffusive dynamics may majorly impact the transport of essential cellular substrates but remains largely uncharacterized. To shed light on the feasibility of resolving spatial from stochastic drivers of diffusive heterogeneity in trajectory data, we developed a framework for predicting our ability to detect differences in diffusivity under different experimental

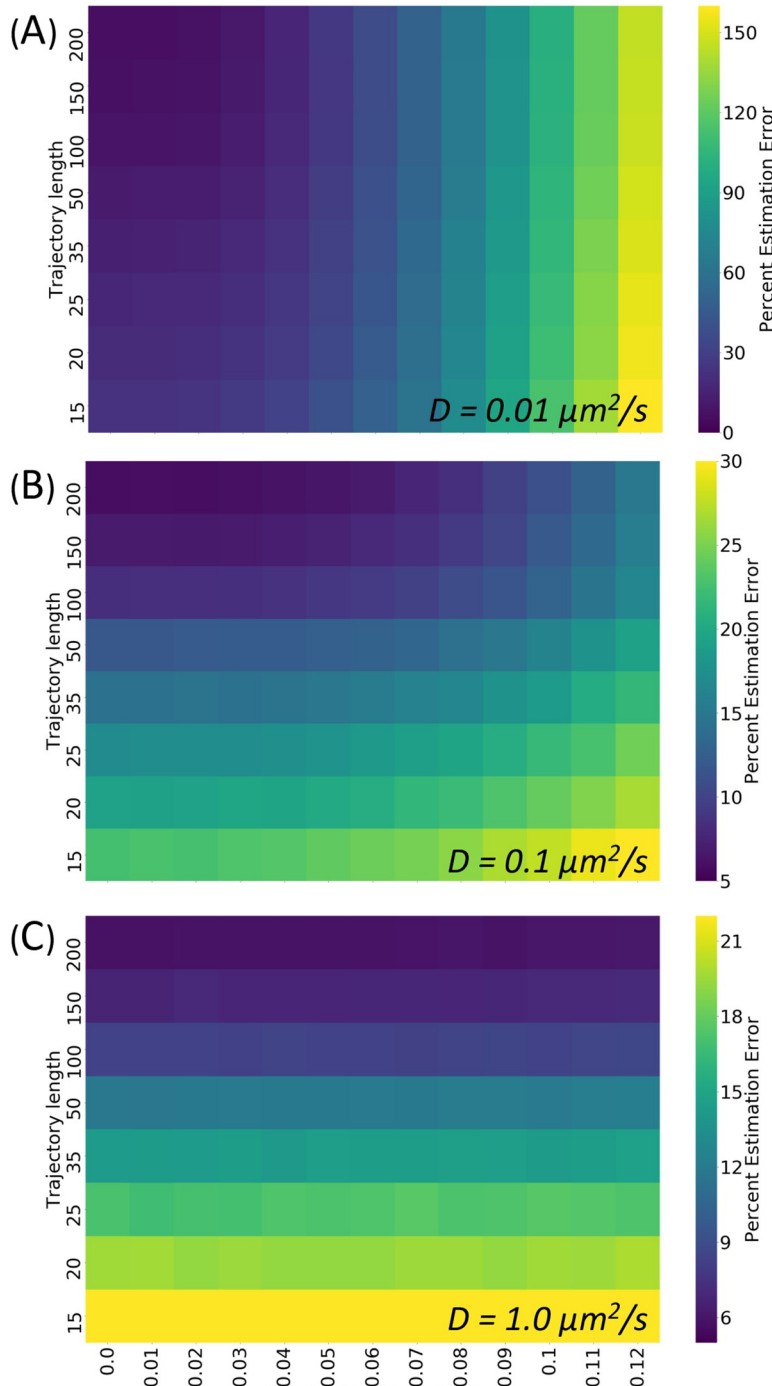

**Fig 7. Percent posterior estimation error conditional on static localization accuracy and trajectory lengths for fraction Brownian motion.** The percent error for a given posterior is measured as the percent error between the true diffusion coefficient used to generate the trajectory, and the mode of the posterior distribution (or the diffusion coefficient which gives the maximum value of the probability density function). Trajectories for this figure are simulated using fractional Brownian motion with Hurst coefficient $H = 0.375$ (or $\alpha = 0.75$) This heatmap reports the mean percent error magnitude for $10^4$ posteriors generated under each set of trajectory length and localization error conditions, with diffusion coefficients of (A) 0.01 $\mu m^2/s$ (B) 0.1 $\mu m^2/s$ and (C) 1.0 $\mu m^2/s$. Please note the difference in heatmap scale bars.

regimes. Our framework is intended to inform the design of experiments characterizing the spatial dependence of diffusivity on sub-cellular location.

## Acknowledgments

We would like to thank Steph Weber, for her helpful comments on this manuscript, and Molly Maleckar, Gabriel Mitchell and Jamie Sherman for their helpful conversations. We thank Jackson Brown for his CookieCutter template and guidance in repository initialization, and Thao Do for her scientific illustration. Finally, we thank Paul G. Allen, founder of the Allen Institute for Cell Science, for his vision, encouragement and support.

## Author Contributions

**Conceptualization:** Julie A. Cass, C. David Williams.

**Data curation:** Julie A. Cass.

**Formal analysis:** Julie A. Cass.

**Investigation:** Julie A. Cass.

**Methodology:** Julie A. Cass, C. David Williams, Julie Theriot.

**Project administration:** Julie A. Cass, C. David Williams, Julie Theriot.

**Resources:** Julie A. Cass, C. David Williams.

**Software:** Julie A. Cass.

**Supervision:** Julie Theriot.

**Validation:** Julie A. Cass.

**Visualization:** Julie A. Cass.

**Writing – original draft:** Julie A. Cass.

**Writing – review & editing:** Julie A. Cass, C. David Williams, Julie Theriot.

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
