## [Decision Letter · Decision Letter 0]

3 Oct 2019

PONE-D-19-23195

A Bayesian framework for the detection of diffusive heterogeneity

PLOS ONE

Dear Dr Cass,

Thank you for submitting your manuscript to PLOS ONE. After careful consideration, we feel that it has merit but does not fully meet PLOS ONE’s publication criteria as it currently stands. Therefore, we invite you to submit a revised version of the manuscript that addresses the points raised during the review process.

We would appreciate receiving your revised manuscript by Nov 17 2019 11:59PM. To enhance the reproducibility of your results, we recommend that if applicable you deposit your laboratory protocols in protocols.io, where a protocol can be assigned its own identifier (DOI) such that it can be cited independently in the future. For instructions see: http://journals.plos.org/plosone/s/submission-guidelines#loc-laboratory-protocols

We look forward to receiving your revised manuscript.

Kind regards,

Juan Carlos del Alamo

Academic Editor

PLOS ONE

**Journal Requirements:**

**Comments to the Author**

1. Is the manuscript technically sound, and do the data support the conclusions?

Reviewer #1: Partly

2. Has the statistical analysis been performed appropriately and rigorously? 

Reviewer #1: No

3. Have the authors made all data underlying the findings in their manuscript fully available?

Reviewer #1: Yes

4. Is the manuscript presented in an intelligible fashion and written in standard English?

Reviewer #1: Yes

5. Review Comments to the Author

Reviewer #1: This study describes a Bayesian inference algorithm to estimate local values of the diffusion coefficient inside live cells from single trajectories of intracellular particles. This type of algorithm can be useful to researchers interested in quantifying diffusivity of heterogeneous or time-varying environments, including but limited to the cytoplasm of live cells. The manuscript describes related existing efforts in the literature, and makes a convincing point that the present algorithm, and in particular its associated freely accessible implementation, is sufficiently different from those existing efforts. In particular, while the parametric nature of the present study is a limitation with respect to existing, non-parametric, efforts, its simplicity may be advantageous to those without significant expertise in statistical mechanics.

The manuscript analyzes the error in the estimated D based on the localization error of the particle. As expected, the error decreases with the length of the recorded trajectory. However, Figure 5 shows this error seems to be unacceptably large for some combinations of values, and it is unclear whether a “typical experiment” (this is a loose term whose meaning is expanded below) would yield acceptable results. The authors argue that the purpose of Figure 5 is for each researcher to assess the error for their own experiments. This is valuable but Figure 5 is plotted in a way that makes this assessment difficult:

1) Only two values of D are covered. It would seem to make more sense to plot Figure 5 normalizing the localization error with (D*tau)^(1/2). This could capture the D-dependence of the estimation error, and only one panel might be necessary to cover all D values.

2) Second, a line plot or contour plot format would be preferable to read errors in the plot.

3) It would be informative to represent a “typical experiment” or experiments in the localization error and trajectory length coordinates of Figure 5. The authors can use experiments from the literature and / or their own data from previous studies.

As the authors point out, a limitation of the study is that it focuses on an idealized model of intracellular diffusion. The authors argue that the method could be adjusted to account for complicated phenomena, such as subdiffusion, but the intended audience of this algorithm may not find this straightforward. This issue is compounded with the fact that this reviewer finds the purely diffusive case to be particularly amenable to the analytical calculation of the posterior distribution. Other cases may be harder… It would be informative to illustrate how the algorithm would be modified in the subdiffusive or e.g., persistent-random walk case by presenting the posterior distribution for those processes.

Finally, there may be cases in which modifying the algorithm to account for non-purely diffusive, isotropic behavior is not feasible or where the actual behavior that needs to be accounted for is unknown a priori. It would be informative to know the error in estimated D in those cases. Again, subdiffusive, persistent-random or anisotropic random walk cases come to mind.

Minor comments:

1) Is the alpha in equation 2 related to the alpha in equation 1?

6. PLOS authors have the option to publish the peer review history of their article (what does this mean?). If published, this will include your full peer review and any attached files.

Reviewer #1: No

---

## [Author Response · Author response to Decision Letter 0]

19 Dec 2019

This response is better viewed in the Response to Reviewers file, where our responses are interwoven with the reviewer's comments and clearly indicated. I have copied this text and placed the reviewer's original notes in brackets, to given our responses appropriate context.

[This study describes a Bayesian inference algorithm to estimate local values of the diffusion coefficient inside live cells from single trajectories of intracellular particles. This type of algorithm can be useful to researchers interested in quantifying diffusivity of heterogeneous or time-varying environments, including but limited to the cytoplasm of live cells. The manuscript describes related existing efforts in the literature, and makes a convincing point that the present algorithm, and in particular its associated freely accessible implementation, is sufficiently different from those existing efforts. In particular, while the parametric nature of the present study is a limitation with respect to existing, non-parametric, efforts, its simplicity may be advantageous to those without significant expertise in statistical mechanics.]

We’d like to thank the reviewer for their thoughtful consideration of our manuscript and helpful comments. Indeed, we hope that this tool may be of particular use to researchers without deeply computational or statistical backgrounds, offering a modifiable tool with more flexibility than the MSD, but less complexity than existing software that can have a higher barrier to entry.

[The manuscript analyzes the error in the estimated D based on the localization error of the particle. As expected, the error decreases with the length of the recorded trajectory. However, Figure 5 shows this error seems to be unacceptably large for some combinations of values, and it is unclear whether a “typical experiment” (this is a loose term whose meaning is expanded below) would yield acceptable results. The authors argue that the purpose of Figure 5 is for each researcher to assess the error for their own experiments. This is valuable but Figure 5 is plotted in a way that makes this assessment difficult:

1) Only two values of D are covered. It would seem to make more sense to plot Figure 5 normalizing the localization error with (D*tau)^(1/2). This could capture the D-dependence of the estimation error, and only one panel might be necessary to cover all D values.

2) Second, a line plot or contour plot format would be preferable to read errors in the plot.]

I agree with the reviewer that this kind of non-dimensionalization would offer a more robust representation of the model results. There are many relevant parameters whose relative values impact the estimation error, and careful choice of how to represent this data is certainly important. However, since we are aiming for this tool to be engaging for a more experimental audience, we wanted to maintain the use of more tangible parameters, which maintain their straightforward physical interpretability, and chose this format to prioritize familiarity and approachability over density of information reporting.

For any given experiment you may have a single localization error and limited range of trajectory lengths. So understand why the parameterization of this figure may not be the most versatile in its use for any one experiment. However, we hope that this parameterization of the lookup table might instead be applicable to a wider audience. As this figure is viewed by a wide audience of researchers with trajectories of varying lengths and localization errors constraining their experiments, we hope they can get a ballpark answer for the question of whether this tool will be useful for them. While the diffusion coefficient value is of course relevant as well, we hope that representing the effects on three different orders of magnitude of diffusivities can give the reader a taste of what might be possible (a third order of magnitude was added in our revised manuscript).

That said, the tool is designed to have an accompanying tutorial Jupyter notebook with examples of how each figure is generated. This was a conscious choice to ensure that those without extensive computational backgrounds have a more easily approached interface with the code, in order to tweak analysis parameters themselves, and see how the results change if they switch out parameter values to generate adaptation of our provided figures which are tailored to their own experimental parameters and constraints. We hope that this design can help make up for the limitations in what can be presented in the manuscript figures.

[3) It would be informative to represent a “typical experiment” or experiments in the localization error and trajectory length coordinates of Figure 5. The authors can use experiments from the literature and / or their own data from previous studies.]

Thank you for this feedback - we have incorporated this suggestion.

[As the authors point out, a limitation of the study is that it focuses on an idealized model of intracellular diffusion. The authors argue that the method could be adjusted to account for complicated phenomena, such as subdiffusion, but the intended audience of this algorithm may not find this straightforward. This issue is compounded with the fact that this reviewer finds the purely diffusive case to be particularly amenable to the analytical calculation of the posterior distribution. Other cases may be harder... It would be informative to illustrate how the algorithm would be modified in the subdiffusive or e.g., persistent-random walk case by presenting the posterior distribution for those processes.

Finally, there may be cases in which modifying the algorithm to account for non-purely diffusive, isotropic behavior is not feasible or where the actual behavior that needs to be accounted for is unknown a priori. It would be informative to know the error in estimated D in those cases. Again, subdiffusive, persistent-random or anisotropic random walk cases come to mind.]

We agree that demonstrating example adaptations to the prior and posterior for more complex intracellular would be an exciting enhancement of what this manuscript could offer, however these advancements are beyond the scope of what we are hoping to explicitly provide within this manuscript. We hope that the detailed demonstration of how this tool is built and acknowledgement that prior and posterior distributions may be adapted for more complex needs is sufficient for demonstrating the value of this tool.

However, we agree with the reviewer’s important note that sub-diffusive motion is of particularly great importance to address in greater detail. To address this important concern, we have taken the reviewers suggestion of applying our analysis tool (with the existing prior and posterior designed for pure diffusion) to biologically- relevant sub-diffusive trajectories and reported the resulting estimation error. For this task we have used trajectories simulated using fractional Brownian motion, as previous work has shown this to be a prevalent mode of intracellular transport. The Hurst coefficient (H) used to define this process is H = alpha/2, where alpha is the parameter giving the MSD’s scaling with time lag (tau) as in Eq 1 of our manuscript. Thus, we have set the Hurst coefficient in the simulated trajectories to best represent reported results for the sub-diffusive time scaling alpha = 0.75 (or H = 0.375).

We feel this was an important addition to the manuscript in demonstrating the applicability of the tool and are grateful for the reviewer’s suggestion to include this.

[Minor comments:

1) Is the alpha in equation 2 related to the alpha in equation 1?]

Thanks for pointing this out! No, the two are not related. We’ve changed the inverse gamma parameters to (a, b) rather than (alpha, beta) to disambiguate.

---

## [Decision Letter · Decision Letter 1]

12 Mar 2020

PONE-D-19-23195R1

A Bayesian framework for the detection of diffusive heterogeneity

PLOS ONE

Dear Dr Cass,

Thank you for submitting your manuscript to PLOS ONE. After careful consideration, we feel that it has merit but does not fully meet PLOS ONE’s publication criteria as it currently stands. Therefore, we invite you to submit a revised version of the manuscript that addresses the points raised during the review process.

We would appreciate receiving your revised manuscript by Apr 26 2020 11:59PM. To enhance the reproducibility of your results, we recommend that if applicable you deposit your laboratory protocols in protocols.io, where a protocol can be assigned its own identifier (DOI) such that it can be cited independently in the future. For instructions see: http://journals.plos.org/plosone/s/submission-guidelines#loc-laboratory-protocols

We look forward to receiving your revised manuscript.

Kind regards,

Juan Carlos del Alamo

Academic Editor

PLOS ONE

Reviewers' comments:

Reviewer's Responses to Questions

**Comments to the Author**

1. If the authors have adequately addressed your comments raised in a previous round of review and you feel that this manuscript is now acceptable for publication, you may indicate that here to bypass the “Comments to the Author” section, enter your conflict of interest statement in the “Confidential to Editor” section, and submit your "Accept" recommendation.

Reviewer #1: (No Response)

Reviewer #2: (No Response)

2. Is the manuscript technically sound, and do the data support the conclusions?

Reviewer #1: Yes

Reviewer #2: Yes

3. Has the statistical analysis been performed appropriately and rigorously? 

Reviewer #1: Yes

Reviewer #2: N/A

4. Have the authors made all data underlying the findings in their manuscript fully available?

Reviewer #1: Yes

Reviewer #2: Yes

5. Is the manuscript presented in an intelligible fashion and written in standard English?

Reviewer #1: Yes

Reviewer #2: Yes

6. Review Comments to the Author

Reviewer #1: I appreciate the authors' efforts to address my concerns and am for the most part satisfied with their revisions. I have a couple of remaining comments.

First, the list of references is rather short and there are places at which the authors discuss standard statistical inference theory without providing appropriate references to the literature. Perhaps a graduate level textbook would be enough. I believe this would be important considering the targeted audience.

Second, I still believe the authors overestimate the generality / flexibility of their approach. I understand the computational framework they present could be extended to other scenarios more representative of intracellular fluctuations than a Gaussian process. However, it is not clear that these extensions would be trivial. In fact, they even recognize this point themselves (circa line 370). I appreciate the authors including a section where they benchmark their tool for fractional Brownian motion. I would suggest to temper the statements about generality of the framework. Also, please use the same color axis and color bars in figures 5 and 7 to facilitate direct comparison (as in by caxis of Matlab or clim of python).

Reviewer #2: The authors manuscript with the accompanying, well-documented python repository is a valuable tool for researchers without significant expertise in Bayesian statistics. It would be more helpful to gather better intuition for KL divergence criterion with more details on how to interpret Fig 6. For e.g, an approximate threshold value of threshold KL below which the posteriors have a given probability to represent the same true diffusion constant (and therefore, not representative of the heterogeneous environment). Authors explain the intuition of KL values, but a rule of thumb would be more beneficial to design experiments.

The authors also provide an accessible way of estimating baseline errors in inference using heatmaps in Fig 5. A very important source of sensitivity to parameter inference lies in prior distribution parameters and a brief guide of choosing parameters (a,b) to not introduce bias in analysis (uninformative prior) would be recommended.

Minor typo: In pg 7/18, the likelihood function is written as p(theta | x) instead of p(x | theta).

7. PLOS authors have the option to publish the peer review history of their article (what does this mean?). If published, this will include your full peer review and any attached files.

Reviewer #1: No

Reviewer #2: No

---

## [Author Response · Author response to Decision Letter 1]

14 Apr 2020

>>>Reviewer #1: I appreciate the authors' efforts to address my concerns and am for the most part satisfied with their revisions. I have a couple of remaining comments.

Thanks for taking the time to consider our manuscript again and for your helpful feedback.

>>>First, the list of references is rather short and there are places at which the authors discuss standard statistical inference theory without providing appropriate references to the literature. Perhaps a graduate level textbook would be enough. I believe this would be important considering the targeted audience.

We appreciate the suggestion and have added references to a graduate Bayesian statistics text (Gelman et al’s “Bayesian Data Analysis”) where appropriate.

>>>Second, I still believe the authors overestimate the generality / flexibility of their approach. I understand the computational framework they present could be extended to other scenarios more representative of intracellular fluctuations than a Gaussian process. However, it is not clear that these extensions would be trivial. In fact, they even recognize this point themselves (circa line 370). I appreciate the authors including a section where they benchmark their tool for fractional Brownian motion. I would suggest to temper the statements about generality of the framework.

We have removed the “flexible” descriptor in line 354 and added another statement acknowledging the challenge of framing more complex priors/ posteriors in line 372-374 in the “Framework limitations” sections.

>>>Also, please use the same color axis and color bars in figures 5 and 7 to facilitate direct comparison (as in by caxis of Matlab or clim of python).

Thanks for catching this; we’ve updated the rendering of these figures so that each comparable plot (ie 5A/7A, 5B/7B etc have identical color axes / color bars.

>>>Reviewer #2: The authors manuscript with the accompanying, well-

documented python repository is a valuable tool for researchers without significant expertise in Bayesian statistics.

We appreciate you taking the time to review our manuscript and accompanying repository.

>>>It would be more helpful to gather better intuition for KL divergence criterion with more details on how to interpret Fig 6. For e.g, an approximate threshold value of threshold KL below which the posteriors have a given probability to represent the same true diffusion constant (and therefore, not representative of the heterogeneous environment). Authors explain the intuition of KL values, but a rule of thumb would be more beneficial to design experiments.

Thanks for this suggestion; we agree that including a benchmark value would increase the usability of this reference table and have included a value and associated brief discussion in lines 332-342.

>>>The authors also provide an accessible way of estimating baseline errors in inference using heatmaps in Fig 5. A very important source of sensitivity to parameter inference lies in prior distribution parameters and a brief guide of choosing parameters (a,b) to not introduce bias in analysis (uninformative prior) would be recommended.

We agree that a discussion of this prior bias and parameter choice strengthens the manuscript and the tool’s usability; we’ve included a brief discussion of this in lines 191-208.

>>>Minor typo: In pg 7/18, the likelihood function is written as p(theta | x) instead of p(x | theta).

Thanks for catching this typo; we’ve fixed it in the updated manuscript.

---

## [Editor Report · Decision Letter 2]

17 Apr 2020

A Bayesian framework for the detection of diffusive heterogeneity

PONE-D-19-23195R2

Dear Dr. Cass,

We are pleased to inform you that your manuscript has been judged scientifically suitable for publication and will be formally accepted for publication once it complies with all outstanding technical requirements.

With kind regards,

Juan Carlos del Alamo

Academic Editor

PLOS ONE
---

## [Editor Report · Acceptance letter]

22 Apr 2020

PONE-D-19-23195R2 

A Bayesian framework for the detection of diffusive heterogeneity 

Dear Dr. Cass:

I am pleased to inform you that your manuscript has been deemed suitable for publication in PLOS ONE. Congratulations! Your manuscript is now with our production department. 

With kind regards,

on behalf of

Dr. Juan Carlos del Alamo 

Academic Editor

PLOS ONE